# A Dynamic Model of Inertia Cone Crusher Using the Discrete Element Method and Multi-Body Dynamics Coupling

**Jiayuan Cheng, Tingzhi Ren \*, Zilong Zhang, Dawei Liu and Xin Jin**

National Engineering Research Center for Equipment and Technology of Cold Rolling Strip, Yanshan University, Qinhuangdao 066004, China; chengjia_yuan@126.com (J.C.); zhangzilong0630@163.com (Z.Z.); liudw@ysu.edu.cn (D.L.); chengjia_yuan@163.com (X.J.)

\* Correspondence: rtz@ysu.edu.cn

**Abstract:** The cone crusher is an indispensable equipment in complex ore mineral processing and a variant of the cone crusher is the inertia cone crusher. A real-time dynamic model based on the multibody dynamic and discrete element method is established to analyze the performance of the inertia cone crusher. This model considers an accurate description of the mechanical motions, the nonlinear contact, and the ore material loading response. Especially the calibration of ore material simulated parameters is based on the Taguchi method for the Design of Experiments. For model verification, the industrial-scale experiment was conducted on a GYP1200 inertia cone crusher. Two different drive speeds were included in the experiments, and the testing devices were used to acquire crusher performances, for instance, displacement amplitude, power draw, product size distribution, and throughput capacity in order to accurately compare simulation results. The preliminary model can be qualitatively evaluated the flow pattern of particles and quantitatively evaluated the crushing force distribution in the concave. Furthermore, the simulation predicts the variety of crusher performances using the drive speed and the fixed cone mass as input variables. The simulation model provides novel insight regarding the improvement of linings wear period, lowering manufacturing cost, and obtaining optimal operation parameters.

**Keywords:** inertia cone crusher; multibody dynamic; discrete element method; simulation; crusher performance

## 1. Introduction

The cone crusher plays a vital role in mineral processing, which has been used widely in medium and fine crushing stages in mineral processing comminution operations [1]. As a variant of the cone crusher, the inertia cone crusher is an extra performance crushing equipment in complex ore mineral processing [2]. The compressive action of the cone inertia crusher is realized by inflicting a rotary swing on the mantle relative to the concave in a certain range of nutation angle. The rotary swing is achieved by the partial block vibrator transferring the rotation of the drive shaft via a connecting shaft to the main shaft, as schematically shown in Figure 1. The concave is placed on several rubber absorbers, which are connected to the ground. Therefore, the concave has 6-DOF in three-dimensional spaces. The mantle can rotate freely along the axis of the main shaft, and the mantle rolls on the bed of particles. The smallest space is defined as the close side setting (CSS), and the largest space between the concave and the mantle is defined as the open side setting (OSS) [3]. Feed particles are usually mixed with hard materials in the inertia cone crusher, where the drive shaft transfers the rotary swing via a flexible connection to the main shaft, thus avoiding the self-locked phenomenon. When the

crusher breaks particles, the CSS of the inertia cone crusher is dynamic, unlike other cone crushers such as hydraulic cone crushers. The CSS may be manipulated to obtain optimum operating parameters, in general, it is affected by the feed, the eccentric static moment, and the eccentric speed.

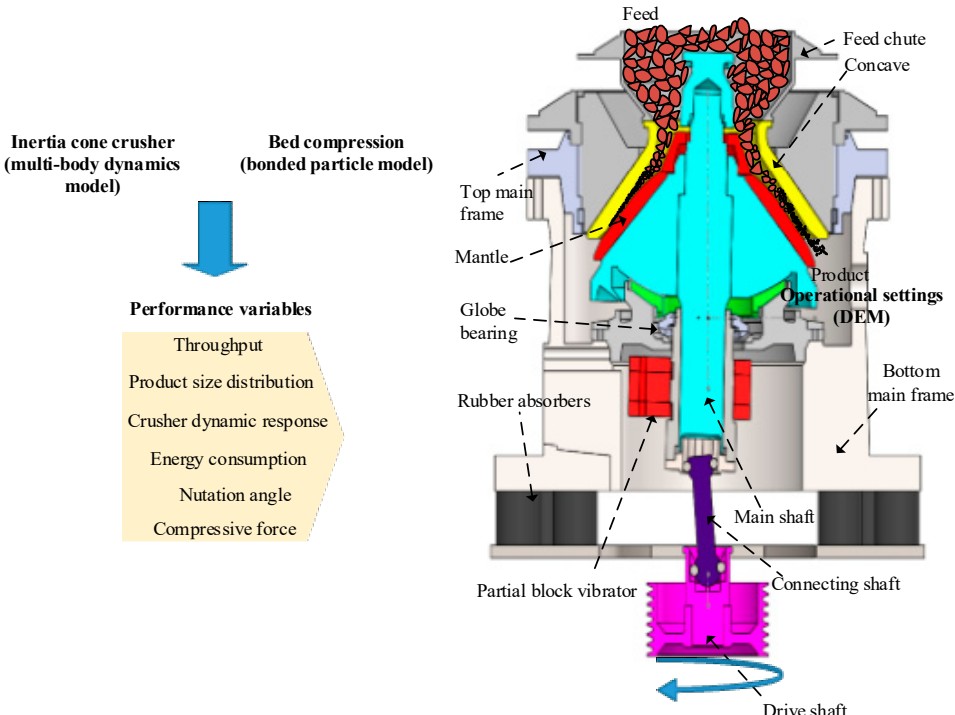

**Figure 1.** Schematics of the inertia cone crusher and the general scheme of the model, as well as its outputs.

An inertia cone crusher operation model of the crushing chamber profile was developed by Babaev et al. [4]. Savov and Nedyalkov [5,6] contributed to crushing force theoretical examination and layer crushing in the inertia cone crusher. Bogdanov et al. [7] described the process of mantle movement on the surface of concave. These above mechanistic models are derived from modal coordinate equations as well as experimental data obtained from industry or laboratory. However, the effect of rock particles on equipment is difficult to consider directly in the models. There are complex contacts between particles and mechanical geometries, which affects the power consumption and dynamic characteristics in inertia cone crushers. It is a very powerful method for a dynamic model of an inertia cone crusher containing particles based on the coupling of multibody dynamics (MBD) and the discrete element method (DEM) for both improving crushing equipment structure and performance. The MBD approach is an effective tool for the dynamics description of multibody systems [8,9]. The DEM [10] is based on molecular dynamics for modeling the flow and mechanical behavior of particles.

Quist and Evertsson [11] modeled a discrete element model of the hydraulic cone crusher using the bonded particle model (BPM) in the commercial software EDEM (2018, DEM Solutions, Edinburgh, Britain), so as to research the influence of the eccentric speed and CSS on product size distribution, hydraulic pressure, and power draw. Li et al. [12], Cleary et al. [13], and Andre et al. [14] investigated the influence of ore material properties and operating parameters on product performance using the particle replacement model (PRM) with spherical and nonspherical particles. These above simulations have been used to describe the effect of the given kinematics of equipment on the product size distribution, output, and the power draw using the DEM. However, these simulations ignore the effect of contacts between the particles and the mechanical geometries on the crusher kinematics, and these simulations will not describe the equipment kinematics which changes according to the particle

behavior characteristics. Currently, no studies have been made to study the dynamics modeling of an inertia cone crusher using the MBD. Furthermore, few scholars have studied the MBD of traditional cone crushers. Several worthwhile attempts had been made to describe the dynamics of vibratory cone crusher. Kazakov and Shishkin [15] modeled the dynamics of vibratory cone crushers using the three-mass system model, ignoring the contact between the particles and the equipment. Jiang et al. [16] equated particle function with hysteretic force and modeled the nonlinear dynamics of vibratory cone crushers using the three-mass system model. Li et al. [17,18] proposed the MBD model of a special 6-DOF robotic cone crusher, ignoring the effect of the equipment kinematics on particle breakage.

Barrios [19] proposed the MBD–DEM primary model of high-pressure grinding rolls (HPGR). Chung [20] established a real-time model based on the coupling of MBD–DEM for a gear in which the holes are filled with elastic particles. These models can be regarded as 2D models, and the coupled MBD–DEM models are not complicated. Currently, no coupled models based on MBD–DEM have been proposed for an inertia cone crusher. This study establishes a real-time model based on coupling of MBD–DEM for an inertia cone crusher in which the crusher chamber is filled with bonded particles.

The MBD analytical approach is used to study high-degree nonlinear problems that cannot be taken as simplified spring–damper models. Connections between the bodies of inertia cone crushers are subject to kinematic constraints, and furthermore, the absolute cartesian coordinate method [21] is used for deriving the kinetic equations for a multibody system. Obviously, the availability of the coupled model depends on the accuracy of the mathematical model. Parameter estimates for the coupled model were set appropriately using calibration experiments. The optimum BPM parameters were obtained so that the critical force and critical compression ratio of the simulated particle are consistent with the actual particle. For model verification, an industrial-scale experiment was conducted, and the results from experiments and simulations were compared. It was able to provide the dynamic response of the crusher, the product performance, and the crushing force within the compressive particles that vary during operation.

## 2. Mathematical Model for the Inertia Cone Crusher with Bonded Particles

The MBD of an inertia cone crusher that interacts with bonded particles is determined by this section, as shown in Figure 2.

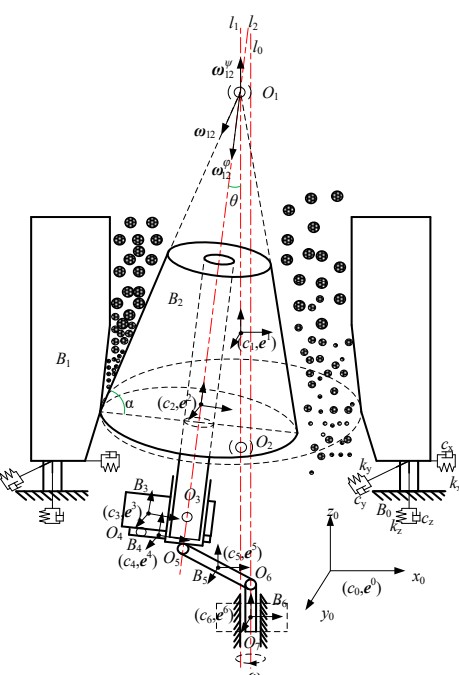

**Figure 2.** Simplified multibody model of an inertia cone crusher with bonded particles.

To simplify the MBD model, the idealizations that are used in the presented paper are as follows:

1.  All components of the inertia cone crusher are taken as rigid bodies.
2.  The drive speed is stable.
3.  The values for the stiffness and the damping for the rubber absorbers are invariant.
4.  The feed particles have the same material properties and uniform sizes.
5.  The particles are uniformly distributed in the chamber which is filled with broken particles.

Based on these assumptions, Figure 2 shows a simplified multibody model of an inertia cone crusher with bonded particles, where $l_0$ is the central axial of the inertia cone crusher, $l_1$ is the central axial of the concave, $l_2$ is the central axial of the mantle, $\theta$ is the nutation angle between the concave and the mantle, $\alpha$ is the structure angle of the mantle. The concave is fixed to the main frame, which is the fixed cone ($B_1$). The mantle is fixed to the main shaft, which is the floating cone ($B_2$). The constraint between the fixed and the floating cone is the spherical joint ($O_1$). The connection between the globe bearing ($B_4$) and the fixed cone is the spherical joint ($O_2$). The constraint between the partial block vibrator ($B_3$) and the floating cone is the cylindrical joint ($O_3$). The connection between the eccentric vibrator and the globe bearing is the planar joint ($O_4$). The constraint between the eccentric vibrator and the connecting shaft ($B_5$) is the ball-pin joint ($O_5$), which constrains two translational motions (x and y) and one rotational motion (Rz) between them. The connection between the connecting shaft and the drive shaft ($B_6$) is the universal joint ($O_6$). The constraint between the drive shaft and ground ($B_0$) is the revolute joint ($O_7$).

### 2.1. Cartesian Coordinate Method Formulation

Seven coordinate frames are used for the dynamic model: one global coordinate frame $(cxyz)_0$ and six local coordinate frames $(cxyz)_i$ ($i = 1, \ldots , 6$), respectively, originate at the center of mass of $B_i$ ($i = 1, \ldots , 6$). The rigid body ($B_i$) in the global reference frame without any constraints has three independent position variables ($x_i, y_i, z_i,$) and three independent angle variables ($\psi_i, \theta_i, \varphi_i$), which are defined as the generalized coordinates of $B_i$. The column matrix $q_i$ for the generalized coordinates of $B_i$ can be expressed as

$$q_i = \left(r_i{}^{\mathrm{T}}, \Lambda_i{}^{\mathrm{T}}\right)^{\mathrm{T}} = (x_i, \ y_i, \ z_i, \ \psi_i, \ \theta_i, \ \varphi_i)^{\mathrm{T}} \tag{1}$$

where T denotes the transposition of a matrix, $r_i$ is a column matrix of position coordinates, and $\Lambda_i$ is a column matrix of Euler angle coordinates. The generalized coordinates of the inertia cone crusher $q$ can be expressed as

$$q = (q_1^{\mathrm{T}}, \ q_2^{\mathrm{T}}, \ q_3^{\mathrm{T}}, \ q_4^{\mathrm{T}}, \ q_5^{\mathrm{T}}, \ q_6^{\mathrm{T}})^{\mathrm{T}} \tag{2}$$

Using Newton–Euler's formulism [22], the equations of dynamic for the rigid body ($B_i$) without any constraints brought on by joints depend on the generalized coordinates and can be expressed as

$$M_i \ddot{q}_i = Q_i \tag{3}$$

where $\ddot{q}_i$ is the column matrix for the generalized accelerations of $B_i$, $M_i$ is the mass matrix for $B_i$, and $Q_i$ is the column matrix for the generalized forces of $B_i$. $M_i$ and $Q_i$ are further expressed as

$$M_i = \begin{bmatrix} m_i E_{3\times3} & 0_{3\times3} \\ 0_{3\times3} & J_i^{(i)} D_i \end{bmatrix}, Q_i = \begin{bmatrix} F_{ia} + F_{if} + F_{ig} + F_{ic} + F_{ip} \\ -\left(J_i^{(i)} \dot{D}_i + \left(D_i \dot{\Lambda}\right) J_i^{(i)} D_i\right) \dot{\Lambda}_i + T_{ia}^{(i)} + T_{if}^{(i)} + T_{ic}^{(i)} + T_{ip}^{(i)} \end{bmatrix} \tag{4}$$

where the superscript (*i*) denotes the coordinates for a vector or matrix in the $B_i$-attached frame $(cxyz)_i$, $m_i$ is the mass of $B_i$, $J_i$ is the inertia matrix for mass center of $B_i$, $E$ denotes the identity matrix, $0$ denotes the null matrix, $F_{ia}$ and $T_{ia}$ are the equivalent absorber force and moment acting on $B_i$, $F_{if}$ and $T_{if}$ are the joint friction force and moment on $B_i$, $F_{ig}$ is the gravity of $B_i$, $F_{ic}$ and $T_{ic}$ are the equivalent

contact force and moment on $B_i$, and $\boldsymbol{F}_{ip}$ and $\boldsymbol{T}_{ip}$ are the equivalent particle forces and torques on $B_i$. The matrix $\boldsymbol{D}_i$ is further expressed as

$$\boldsymbol{D}_i = \begin{bmatrix} \sin\theta_i\sin\varphi_i & \cos\varphi_i & 0 \\ \sin\theta_i\cos\varphi_i & -\sin\varphi_i & 0 \\ \cos\theta_i & 0 & 1 \end{bmatrix} \tag{5}$$

The equations of dynamic for the inertia cone crusher without any constraints can be expressed as

$$\begin{cases} \boldsymbol{M\ddot{q}} = \boldsymbol{Q} \\ \boldsymbol{M} = \mathrm{diag}(\boldsymbol{M}_1,\ \boldsymbol{M}_2,\ \boldsymbol{M}_3,\ \boldsymbol{M}_4,\ \boldsymbol{M}_5,\ \boldsymbol{M}_6) \\ \boldsymbol{Q} = \left(\boldsymbol{Q}_1^{\mathrm{T}},\ \boldsymbol{Q}_2^{\mathrm{T}},\ \boldsymbol{Q}_3^{\mathrm{T}},\ \boldsymbol{Q}_4^{\mathrm{T}},\ \boldsymbol{Q}_5^{\mathrm{T}},\ \boldsymbol{Q}_6^{\mathrm{T}}\right)^{\mathrm{T}} \end{cases} \tag{6}$$

where $\ddot{\boldsymbol{q}}$ is the matrix for the generalized acceleration coordinates, $\boldsymbol{M}$ is the mass matrix for the system, and $\boldsymbol{Q}$ is the matrix for the generalized forces of the system.

The holonomic constraint is defined as a set of constraint equations with only displacement coordinates. Such holonomic constraints brought on by joints $O_j$ ($j$ = 1, 2, 3, 4, 5, 6, 7) of the multibody system and driving motion constraints of the drive shaft are described by holonomic constraint equations as

$$\boldsymbol{\Phi}(\boldsymbol{q},t) = 0,\ \boldsymbol{\Phi}_q = \frac{\partial\boldsymbol{\Phi}}{\partial\boldsymbol{q}} = \left[\frac{\partial\Phi_i}{\partial q_j}\right]; i = 1,\ldots\ldots,26,\ j = 1,\ldots\ldots,7 \tag{7}$$

Equation (7) gives the velocity and acceleration relation and can be expressed as

$$\begin{cases} \boldsymbol{\Phi}_q\dot{\boldsymbol{q}} = -\boldsymbol{\Phi}_t \\ \boldsymbol{\Phi}_q\ddot{\boldsymbol{q}} = \zeta \end{cases},\ \zeta = -\left[\left(\boldsymbol{\Phi}_q\dot{\boldsymbol{q}}\right)_q\dot{\boldsymbol{q}} + 2\boldsymbol{\Phi}_{qt}\dot{\boldsymbol{q}} + \boldsymbol{\Phi}_{tt}\right] \tag{8}$$

where $\boldsymbol{\Phi}_t = \partial\boldsymbol{\Phi}/\partial t$, the Jacobian matrix for $\boldsymbol{\Phi}_q\dot{\boldsymbol{q}}$ is $\left(\boldsymbol{\Phi}_q\dot{\boldsymbol{q}}\right)$, $\boldsymbol{\Phi}_{qt} = \partial\boldsymbol{\Phi}_q/\partial t$, $\boldsymbol{\Phi}_{tt} = \partial\boldsymbol{\Phi}_t/\partial t$.

Based on the Lagrange multiplier formulation and kinematically admissible virtual displacements of Equation (8), the equations of dynamic for the inertia cone crusher are obtained by the generalized system coordinates and the Lagrange multipliers, and can be expressed as

$$\begin{bmatrix} \boldsymbol{M} & \boldsymbol{\Phi}_q^{\mathrm{T}} \\ \boldsymbol{\Phi}_q & 0 \end{bmatrix}\begin{bmatrix} \ddot{\boldsymbol{q}} \\ \lambda \end{bmatrix} = \begin{bmatrix} \boldsymbol{Q} \\ \zeta \end{bmatrix},\ \lambda = (\lambda_1,\lambda_2,\cdots,\lambda_{26})^{\mathrm{T}} \tag{9}$$

*2.2. Nonlinear Contact Model*

In the case of the normal operation of the inertia cone crusher, the mantle and the concave will eventually come into contact or collide and exert forces between them. The inertia cone crusher has a nonlinear contact problem, unlike hydraulic cone crushers. An instance of the collision between the fixed cone and floating cone is shown in Figure 3. The contact force $\boldsymbol{F}_{ic}$ [23,24] is calculated as

$$\boldsymbol{F}_{ic} = \boldsymbol{f}_n + \boldsymbol{f}_t \tag{10}$$

where $\boldsymbol{f}_n$ and $\boldsymbol{f}_t$ indicate normal constraint force and friction force, respectively. If the indentation is written as $\delta$, then the value of normal force becomes

$$\left|\boldsymbol{f}_n\right| = k\delta^{h_1} + c\frac{\dot{\delta}}{\left|\dot{\delta}\right|}\left|\dot{\delta}\right|^{h_2}\delta^{h_3} \tag{11}$$

where $k$ is the stiffness coefficient, $c$ is the damping coefficient, $h_1$ and $h_2$ are exponents for the stiffness and the damping, and $h_3$ is the indentation exponent.

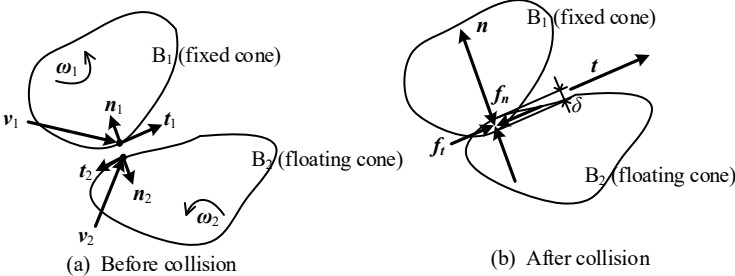

**Figure 3.** Collision of the fixed cone and floating cone.

The spring coefficient $k$ varies with the material and shape of the two cones, and the relative velocity at the initial point of contact. Hertz [25] theoretically derived that the spring coefficient $k$ varies between two spheres, expressed as

$$k = \frac{4}{3\pi} \frac{1}{\left(\frac{1}{k_1} + \frac{1}{k_2}\right)\sqrt{\frac{1}{r_1} + \frac{1}{r_2}}}, \; k_i = \frac{\pi E_i}{1 - v_i^2} \; (i = 1, \, 2) \tag{12}$$

where $E_i$, $r_i$, and $v_i$ are the Young's modulus, radius, and Poisson's ratio for the $i$ sphere. The value of the damping coefficient $c$ also has unknown portions, so theoretical evaluations for this value are also difficult. For a tentative estimation of $c$, $k$ value of 1/1000 to 1/10,000 is often used if the "kg, s, mm, N" unit system is used.

Let $t$ be a unit vector tangent to the sliding plane between two cones, and $v_1$ and $v_2$ be the absolute velocity vectors at a contact point on the fixed cone ($B_1$) and the floating cone ($B_2$). Then, the value of friction $f$ and $v$ the velocity of $B_2$ with respect to $B_1$ are given as

$$\left|f_t\right| = \mu(v)\left|f_n\right|, \; v = (v_2 - v_1) \cdot t \tag{13}$$

where $\mu(v)$ is the coefficient with sign $v_1$ and $v_2$ and $t$ are expressed in vector form with respect to the global coordinate frame $(cxyz)_0$.

The "Friction Coefficient" is given by using "Parametric Values". This is the case when only slip motion is taken into account. The friction coefficient $\mu(v)$ is defined by Equation (14) [26].

$$\mu(v) = \begin{cases} \mu_d & ; & v \leq -v_d \\ \text{hav}\sin(v, -v_d, \mu_d, -v_s, \mu_s) & ; & -v_d < v \leq -v_s \\ \text{hav}\sin(v, -v_s, \mu_s, v_s, -\mu_s) & ; & -v_s < v < v_s \\ \text{hav}\sin(v, v_s, -\mu_s, v_s, -\mu_d) & ; & v_s \leq v < v_d \\ -\mu_d & ; & v_d \leq v \end{cases} \tag{14}$$

where $\mu_d$ is the dynamic friction threshold coefficient, $\mu_s$ indicates the static friction threshold coefficient, $v_d$ is the dynamic threshold velocity, $v_s$ is the static threshold velocity, and $v$ is the tangential relative velocity. The havsin $(x, x_0, h_0, x_1, h_1)$ can be expressed as

$$\text{hav}\sin(x, x_0, h_0, x_1, h_1) = \begin{cases} h_0 & ; & x < x_0 \\ \frac{h_0 + h_1}{2} + \frac{h_1 - h_0}{2}\sin\left(\frac{x - x_0}{x_1 - x_0}\pi - \frac{\pi}{2}\right) & ; & x_0 \leq x \leq x_1 \\ h_1 & ; & x_1 \leq x \end{cases} \tag{15}$$

In this study, an industry-scale inertia cone crusher (GYP1200 type, Beijing Kaite Company, Beijing, China) is simulated and the MBD for the inertia cone crusher without any particles is modeled using the software RecurDyn. Some important simulation parameters for the RecurDyn are presented in Table 1.

**Table 1.** Parameters for the RecurDyn simulation.

| Parameter | Value | Unit |
|---|---|---|
| Machine | | |
| Crusher model | GYP1200 | - |
| Mantle cone angle | 50 | (deg) |
| Eccentric vibrator speed | (400,600) | (rpm) |
| Fixed cone mass | 24,124.4 | (kg) |
| Floating cone mass | 4495.1 | (kg) |
| Eccentric vibrator mass | 784.5 | (kg) |
| Rubber absorber properties | | |
| Spring coefficient ($k_x, k_y, k_z$) | (320,320,970) | (N/mm) |
| Damping coefficient ($c_x, c_y, c_z$) | (100,100,100) | (N·s/mm) |
| Nonlinear contact parameters | | |
| Stiffness coefficient | $10^5$ | (N/mm) |
| Damping coefficient | 50 | (N·s/mm) |
| Stiffness exponent | 1.5 | - |
| Damping exponent | 1 | - |
| Indentation exponent | 2 | - |
| Dynamic threshold coefficient | 0.25 | - |
| Static threshold coefficient | 0.3 | - |
| Dynamic threshold velocity | 10 | (mm/s) |
| Static threshold velocity | 1 | (mm/s) |

## 2.3. Particle Compression Model Using DEM

DEM can solve the kinetic equation for each particle using a step-by-step numerical integration. Two different models are available for the DEM simulation. The Hertz-Mindlin contact model [27,28] is available for the dynamic behavior between particles and geometries or between particles. The bonded particle model (BPM) [29,30] is used to model a particle cluster which has tetrahedral mesh cells with damping contacts and stress restrictions in both normal and tangential direction. The Hertz-Mindlin contact model is used for a spring–damper model in the normal and tensile directions, and a frictional slider in the tensile direction, as shown in Figure 4. The contact force is made up of the tangential part and the normal part $F_{pn}$ which is expressed as

$$F_{pn} = F_{pn,s} + F_{pn,d}, \ F_{pn,s} = \frac{4}{3} E_p^* \sqrt{r^*} \delta_n^{3/2}, \ F_{pn,d} = -2\sqrt{\frac{5}{6}} \xi_d \sqrt{M^* k_n} \dot{\delta}_n \tag{16}$$

where $\delta_n$ is the contact overlap in the normal direction, the normal contact stiffness $k_n$ is expressed as $k_n = 2E_p^* \sqrt{r^* \delta_n}$, $e$ is the restitution coefficient, and $\xi_d = -\ln e / \sqrt{(\ln e)^2 + \pi^2}$. The generalized Young's modulus $E_p^*$, the generalized mass $M^*$ and radius $r^*$ are expressed in Equation (17).

$$E^* = \left( \frac{1 - \left(v_p^{(i)}\right)^2}{E_p^{(i)}} + \frac{1 - \left(v_p^{(j)}\right)^2}{E_p^{(j)}} \right)^{-1}, \ M^* = \left( \frac{1}{m^{(i)}} + \frac{1}{m^{(j)}} \right)^{-1}, \ r^* = \left( \frac{1}{r^{(i)}} + \frac{1}{r^{(j)}} \right)^{-1} \tag{17}$$

where $E_p^{(i)}$ and $E_p^{(j)}$, respectively, denote the Young's modulus for the particle i, j. $v_p^{(i)}$ and $v_p^{(j)}$ respectively, are the Poisson's ratio for the particle *i. j*; $m^{(i)}$ and $m^{(j)}$ are the masses of the particle *i, j*. $r^{(i)}$ is the radius of the particle *i*. $r^{(j)}$ is the radius of the particle *j*.

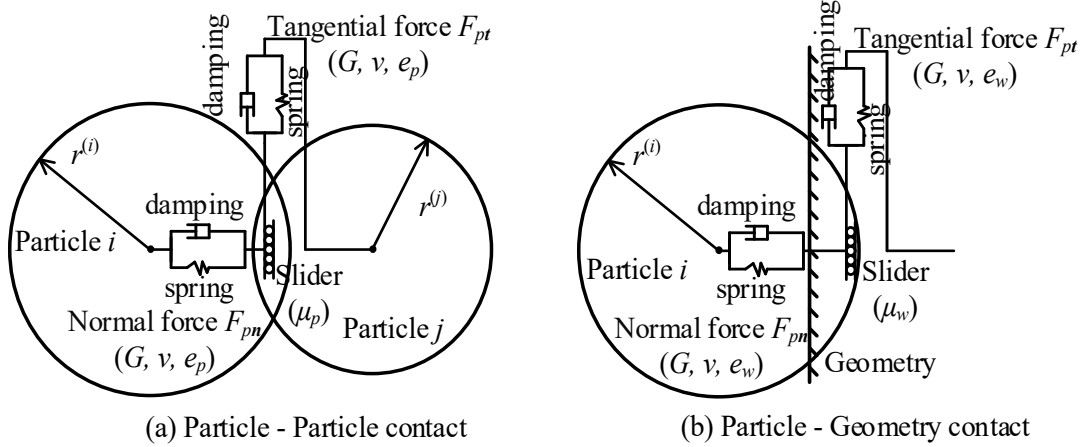

**Figure 4.** A damped Hertz-Mindlin contact model.

The tangential component of the contact force $F_{pt}$ can be expressed as

$$F_{pt} = F_{pt,s} + F_{pt,d},\ F_{pt,s} = -k_t\delta_t,\ F_{pt,d} = -2\sqrt{\frac{5}{6}}\xi_d\sqrt{M^*k_t}\dot{\delta}_t \tag{18}$$

where $\delta_t$ is the contact overlap in the tangential direction, and the tangential contact stiffness $k_t$ is expressed as $k_t = 8G_p^*\sqrt{r^*\delta_n}$, $G_p^*$ is the equivalent shear modulus. The tangential contact force is determined by a Coulomb friction law expressed as

$$|F_{pt}| \le \mu|F_{pn}| \tag{19}$$

where $\mu$ is the friction coefficient. For the Hertz-Mindlin contact model. The contact friction and damping coefficient are two important ways for the energy consumption of the system of the particle assembly. The necessary parameters are Poisson's ratio $v$, shear modulus $G$, the friction coefficient between particles and geometries $\mu_w$, the friction coefficient between particles $\mu_p$, the restitution coefficient between particles and geometries $e_w$, and the restitution coefficient between particles $e_p$.

The bonded particle model is used to bond a particle cluster together forming a breakage body. The particle forming a cluster is defined as a metaparticle, and the cluster created is defined as a breakage particle. For metaparticles in contact, a virtual beam will be created, as shown in Figure 5. In each bonded contact, the virtual beam of the force–displacement characteristics is determined by the five parameters that define a theoretical bond: the normal stiffnesses per unit area $k_{bn}$, the shear stiffnesses per unit area $k_{bt}$, the tensile strength $\sigma_{bc}$, the shear strength $\tau_{bc}$, the bond radius multiplier $\lambda_b$. The theoretical bond radius $R_b$ can be expressed as

$$R_b = \lambda_b\min\left(r^{(i)}, r^{(j)}\right) \tag{20}$$

where $r^{(i)}$ and $r^{(j)}$, respectively, are the radius of particle $i, j$. The theoretical bonds can transfer both force and torque between particles, while particles can only transfer force, and these springs are defined as a virtual beam whose length $L_b$.

The $F_b$ and $T_b$, respectively, denotes the force and torque carried by the theoretical bond expressed as

$$F_b = F_{bn} + F_{bt},\ T_b = T_{bn} + T_{bt} \tag{21}$$

where $F_{bn}$, $F_{bt}$ and $T_{bn}$, $T_{bt}$ denote the forces and torques in the normal and tangential direction, respectively. The interaction between particles is based on the Hertz-Mindlin contact model and $F_b$ and $T_b$ are set to zero before the theoretical bond formation or after breakage. After bond formation, each relative displacement and angle increment ($\Delta U_n$, $\Delta U_t$, $\Delta \Theta_n$, $\Delta \Theta_t$) forms an increment of spring

force and torque which is set to the current values. The force or torque is derived according to Equation (22).

$$\delta\boldsymbol{F_{bn}} = -k_{bn}A\Delta\boldsymbol{U_n},\ \delta\boldsymbol{F_{bt}} = -k_{bt}A\Delta\boldsymbol{U_t},\ \delta\boldsymbol{T_{bn}} = -k_{bn}J\Delta\boldsymbol{\Theta_n},\ \delta\boldsymbol{T_{bt}} = -k_{bt}\frac{J}{2}\Delta\boldsymbol{\Theta_t} \tag{22}$$

where $A$ is the area and $J$ is the polar moment of inertia for the theoretical bond. These increments are expressed as

$$\Delta\boldsymbol{U_n} = \boldsymbol{v_n}\delta t,\ \Delta\boldsymbol{U_t} = \boldsymbol{v_t}\delta t,\ A = \pi R_b^2,\ \Delta\boldsymbol{\Theta_n} = \boldsymbol{\omega_n}\delta t,\ \Delta\boldsymbol{\Theta_t} = \boldsymbol{\omega_t}\delta t,\ J = \frac{1}{2}\pi R_b^4 \tag{23}$$

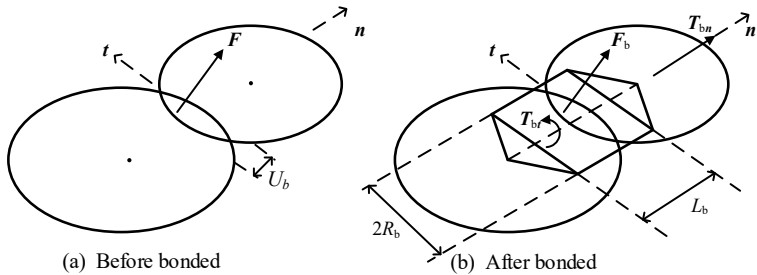

(a) Before bonded          (b) After bonded

**Figure 5.** Force–displacement model of particle bonds.

The maximum tangential stress or normal stress acting on the theoretical bond periphery are derived according to Equation (24).

$$\sigma_b^{max} < \frac{-F_{bn}}{A} + \frac{2|T_{bt}|}{J}R_b,\ \tau_b^{max} < \frac{|F_{bt}|}{A} + \frac{|T_{bn}|}{J}R_b \tag{24}$$

When the maximum normal stress or tangential stress exceeds the tensile strength or the shear strength, as expressed in Equation (25), the theoretical bond breaks.

$$\sigma_b^{max} \geq \sigma_{bc},\ \tau_b^{max} \geq \tau_{bc} \tag{25}$$

### 2.4. The Bonded Particle Model Calibration

Obviously, the key to using DEM to simulate rock breakage depends on the accuracy of the BPM which corresponds to realistic breakage behavior. If the BPM parameters are improper the simulated particle will be fractured in an inaccurate behavior, so it is essential to calibrate the BPM parameters. Firstly, a series of Brazilian strength tests and single-particle compressive breakage tests were conducted in order to log the critical force and critical compression ratio for the breakage and estimate the strength of the rock. Secondly, the breakage simulation was established using the Taguchi method for the Design of Experiments (DOE). Finally, the difference between the data collected and results simulated was minimized by the optimum parameters form the BPM calibration.

#### 2.4.1. A Bimodal Particle Packing Cluster

The packing density within the breakage particle increases, which is based on a widely bimodal distribution [31,32]. The particle size distribution of the metaparticles forming the breakage particle needs to be chosen so that breakage is properly described while keeping the computational economy, and three different packing types can be shown in Figure 6. The best possibilities for gaining good breakage characteristics and packing density using the least particles are controlled by the bimodal distribution which uses smaller particles to conglutinate relatively large particles in the coarse end. In this study, the rock material is white marble, so the coarse end of the bimodal distribution has a mean

radius of 3 mm and a standard deviation of 0.525 mm. The fine end of the bimodal distribution has a mean radius of 1 mm and a standard deviation of 0.225 mm. The packing density of the metaparticle bed is about 0.82. In the inertia cone crusher simulation, a 3D model of rocks is used as the breakage particle model, as shown in Figure 7. In this work, four different simulated particles were used to correspond to the feed size distribution in the industrial-scale experiment.

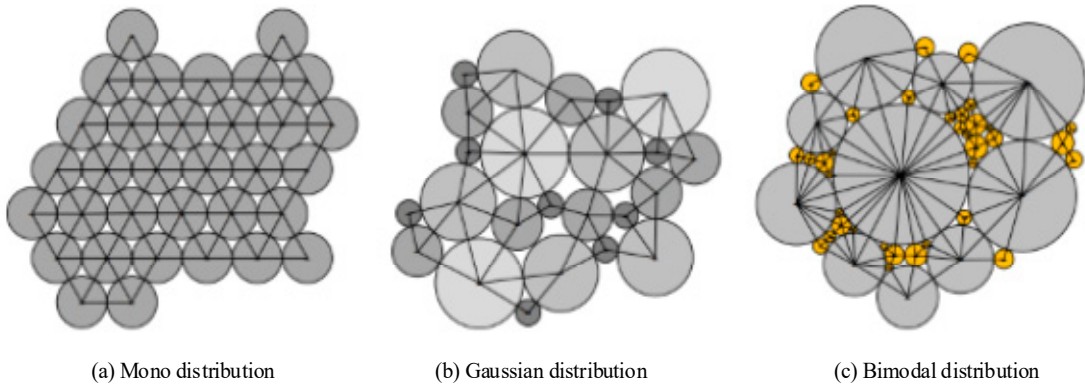

(a) Mono distribution      (b) Gaussian distribution      (c) Bimodal distribution

**Figure 6.** Three different packing types determined by different manners of particle size distributions.

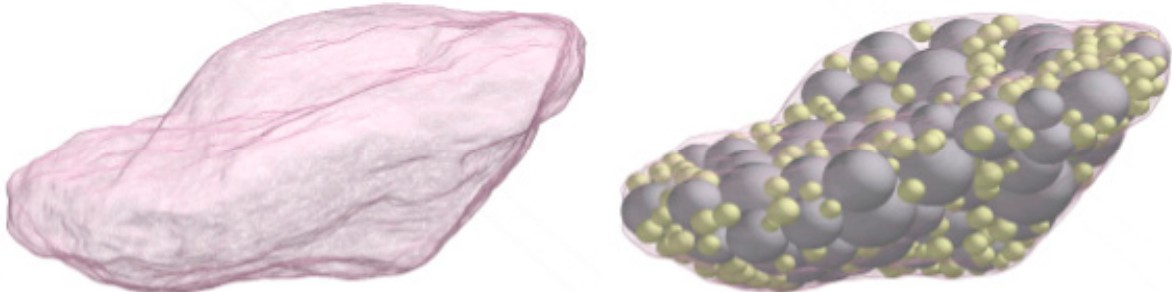

**Figure 7.** A 3D scanned rock model is used for forming breakage particles with real shape.

2.4.2. Laboratory Breakage Experiments

A set of Brazilian tests [33] and a single-particle crushing test of white marble samples were conducted, and the differences of samples lead to a difference in the critical compression ratio and force needed to break rock samples of cylindrical shape, as shown in Figure 8. The diameter of the sample disks is 50 mm and the height is 50 mm. The samples of cylindrical shape are compressed by 3%, which normally can break the samples. The force-displacement curves for the six performed Brazilian tests are shown in Figure 9. It can be seen that the force increases until the sample breakage, so when breakage occurs, the peak force is defined as the critical force, and the compression ratio is defined as the critical compression ratio.

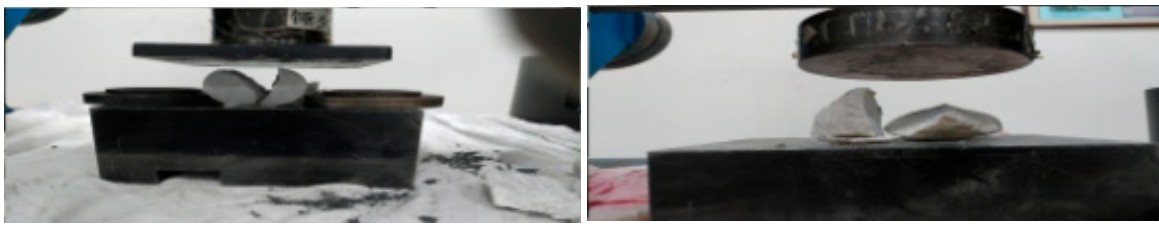

(a) Brazilian splitting tensile strength test      (b) Single particle crushing test

**Figure 8.** Photo of the crushing test on a white marble particle from the feed sample.

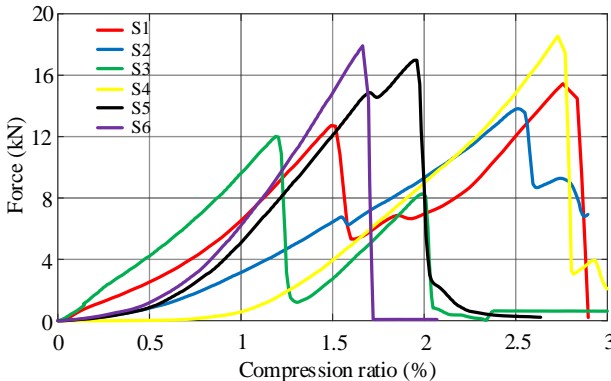

**Figure 9.** The figure shows the force-displacement curves for six Brazilian tests.

An average value of 14.8 kN for the critical force was calculated from the data, and the average strength was 3.77 MPa according to Equation (26)

$$\sigma_t = 2F_c/\pi DL \tag{26}$$

where $F_c$ is the critical force, $D$ is the sample diameter, and $L$ is the sample thickness.

### 2.4.3. Breakage Simulations Established Using DOE

The mechanics of the simulated particle response under Brazilian tests are often complex and nonlinear. Therefore, the critical force and the critical compression ratio were selected for evaluating responses. A rock from the cylindrical shape simulation test was used for the Brazilian splitting tensile strength test, as shown in Figure 10. The bond and force structure of the breakage particle is displayed, and broken bonds are shown in black. The calibration of the BPM parameters is described by simulations of 16 different factor combinations. Hanley et al. [34] and Yoon [35] calibrated the BPM parameters using the Taguchi method [36] for DOE.

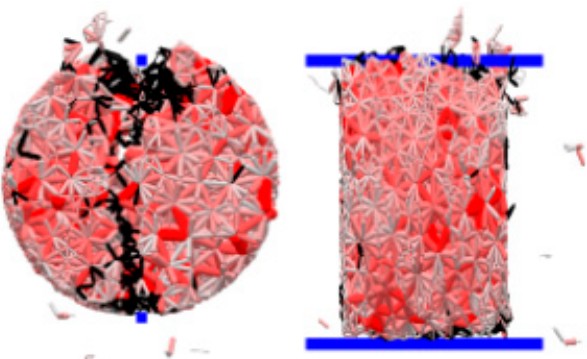

**Figure 10.** The picture shows the breakage process Brazilian test in a BPM calibration simulation.

Table 2 shows the parameters (factors) and their settings (levels) in the Brazilian test simulations. An L16 ($4^5$) array had been chosen, as it can contain five factors (including an error factor) at four levels. A stiffness ratio $k_{bn}/k_{bt}$ is set to 2.5 according to Wang [37].

**Table 2.** The standard L16 array allocated to each parameter.

| Control Factors | Levels | | | | Column Assigned |
|---|---|---|---|---|---|
| | 1 | 2 | 3 | 4 | |
| Normal critical stress (MPa) | 22 | 17 | 25 | 20 | 1 |
| Shear critical stress (MPa) | 16 | 13 | 9 | 11 | 2 |
| Shear stiffness (GPa/m) | 100 | 120 | 160 | 200 | 4 |
| Bond disk radius (mm) | 3.2 | 3.0 | 3.3 | 3.1 | 5 |
| Error - | | | | | 3 |

2.4.4. The Optimum BPM Parameters

The data were analyzed using Minitab 16. An ANOVA was used for the original data for each response, as shown in Table 3. The force difference $F_{cd}$ is defined as

$$F_{cd} = |F_{cs} - F_{cav}| \tag{27}$$

where $F_{cs}$ is the critical force from the simulation and $F_{cav}$ is the average critical force from the experiment.

**Table 3.** ANOVA results of the two responses: *DF* is the number of free degrees, *SS* the sum of standard squares, *MS* the mean square, and is *F* the *f*-value calculated with respective variances.

| Force Difference | | | | | Critical Compression Ratio | | | | |
|---|---|---|---|---|---|---|---|---|---|
| Factor | DF | SS | MS | P | Factor | DF | SS | MS | P |
| Shear critical stress | 3 | 72.29 | 24.10 | *0.133* | Normal critical stress | 3 | 22.348 | 7.449 | *0.185* |
| Shear stiffness | 3 | 61.83 | 20.61 | *0.125* | Shear critical stress | 3 | 20.041 | 9.680 | *0.174* |
| Bond disk radius | 3 | 18.46 | 6.15 | 0.001 | Shear stiffness | 3 | 12.051 | 4.017 | 0.002 |
| Normal critical stress  Equivalent errors | 3  6 | 5.64  13.06 | 1.88  4.35 | | Bond disk radius | 3 | 20.135 | 6.717 | *0.102* |
| Error | 3 | 7.42  2.47 | | | Error | 3 | 3.653 | 1.218 | |
| Total | 15 | 165.64 | | | Total | 15 | | | |

In this ANOVA, the force difference $F_{cd}$ and the critical compression ratio are selected as evaluating responses. This section evaluates the statistical significance using the *p*-value: If the *p*-values exceed 0.1, the effect can be seen as significant at the 0.1 significance level. In order to better visualize, using bold and italics show statistically significant effects. It is clear that the most influential parameters for the force difference are shear critical stress and shear stiffness. For the critical compression ratio, the most influential parameters are shear critical stress and normal critical stress. The bond disk radius is also the major factor for the critical compression ratio at the 0.1 significance level. Figure 11 shows the marginal means graphs for two responses for the simulations in ANOVA.

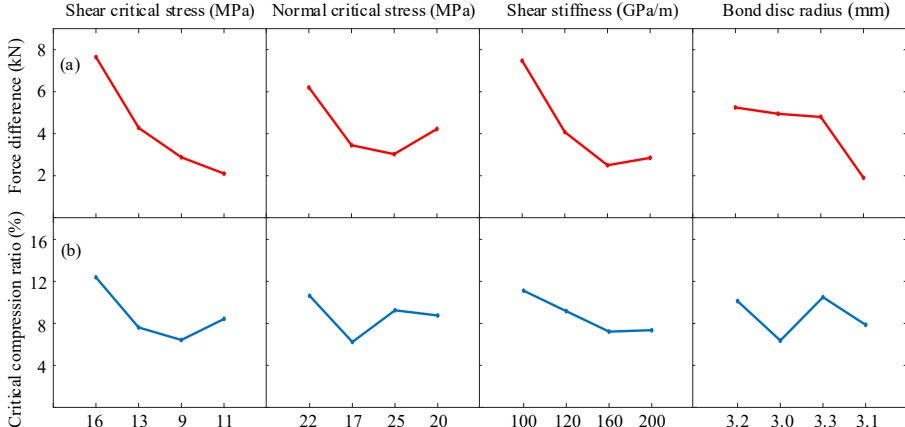

**Figure 11.** Marginal means graphs for two responses for the Brazilian test simulations: (**a**) force difference; (**b**) critical compression ratio.

The optimum BPM parameters are obtained from a comprehensive analysis of view: shear critical stress 9 MPa, normal critical stress 25 MPa, shear stiffness 160 GPa/m, and bond disk radius 3.1 mm. A set of single-particle compressive breakage experiments and simulations of samples in four different sizes and shapes were conducted. The optimum BPM parameters were used in single-particle compressive breakage simulations, and the force-displacement curves for tests and simulations can be shown in Figure 12.

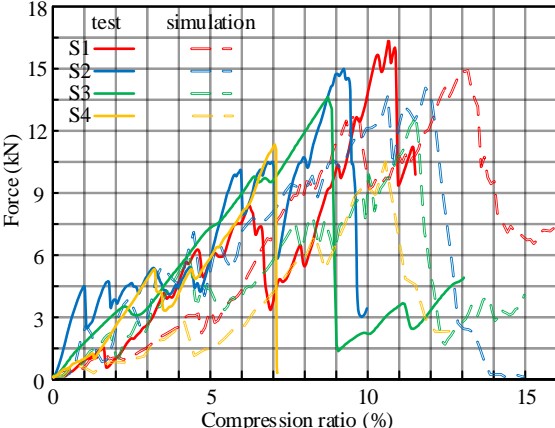

**Figure 12.** The force-displacement curves for the four performed single-particle breakage tests.

In terms of the critical force. Figure 12 shows how the simulation results correspond well to the test data, whereas in terms of the critical compression ratio, there are some differences between the simulation results and the test data. The critical compression ratio of the simulated particle is not more than 1.5 times of the actual particle. For obtaining the appropriate compression ratio, the particles and bond stiffness will be set high, so the simulation time steps need to be very small. Clearly, this is not feasible. In this study, DEM simulations were conducted using the software EDEM. Some important simulation parameters for the EDEM are presented in Table 4.

**Table 4.** Parameters for the EDEM simulation.

| Parameter | Value | | Unit |
|---|---|---|---|
| DEM material properties | | | |
| | Rock | Steel | |
| Density | 2600 | 7800 | (kg/m$^3$) |
| Shear stiffness | $5.08 \cdot 10^8$ | $7.0 \cdot 10^{10}$ | (Pa) |
| Poisson's ratio | 0.35 | 0.3 | - |
| | Rock-Rock | Rock-Steel | |
| Static friction coefficient | 0.5 | 0.7 | - |
| Restitution coefficient | 0.15 | 0.2 | - |
| Rolling friction coefficient | 0.001 | 0.001 | - |
| BPM parameters | | | |
| Normal stiffness | 400 | | (GPa/m) |
| Shear stiffness | 160 | | (GPa/m) |
| Normal critical stress | 25 | | (MPa) |
| Shear critical stress | 9 | | (MPa) |
| Bond disk radius | 3.1 | | (mm) |
| Simulation | | | |
| Time step | $1.3 \cdot 10^{-7}$ | | (s) |
| Frequency | 1000 | | (Hz) |
| Number of particles | 229,409 | | - |
| Simulation time (400 rpm) | 1080 | | (CPUH) |
| Simulation time (600 rpm) | 1370 | | (CPUH) |
| CPU clock frequency | 4.32 | | (GHz) |
| CPU cores | 24 | | - |

## 3. Simulation Scheme for the Coupled Model and Validation for Industrial Experiments

### 3.1. Numerical Calculation Flow Chart for Coupling of MBD–DEM

To solve Equation (9), the numerical calculation flow chart is shown in Figure 13. Before solving, the material properties and size of mechanical geometries and particles are imported. Firstly, the DEM loop is calculated, which determines the position and velocity, forces, and geometries for each particle based on the Hertz-Mindlin contact force model and the bonded particle model. After the end of the DEM loop, the DEM transmits the *e* forces and torques on the geometries to the MBD loop. The MBD loop solves the differential algebra equations for the MBD, and calculates the positions and velocities of the geometries. When each MBD loop is computed, the corresponding motions of the geometries are transmitted to the DEM loop.

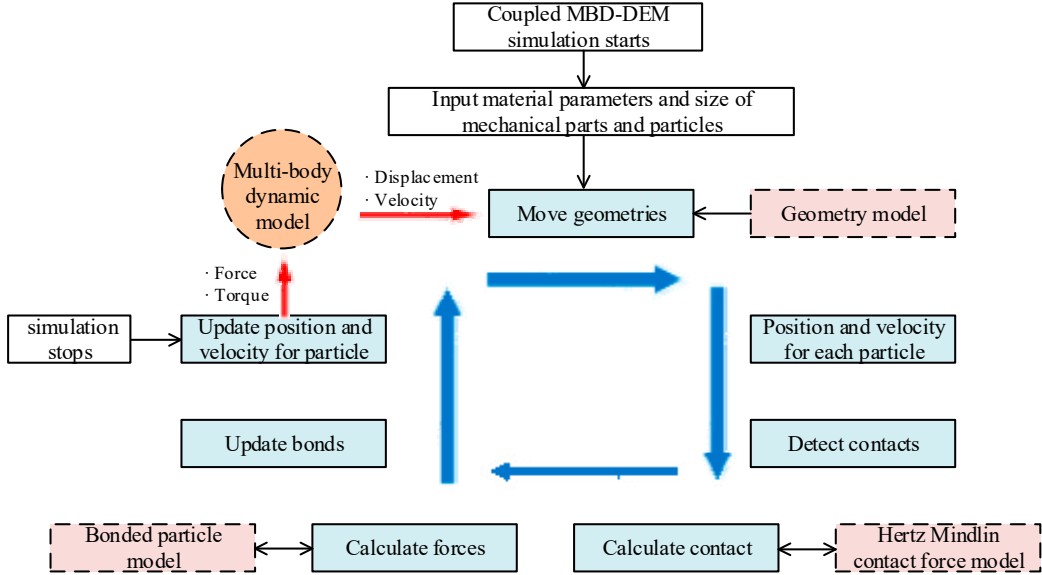

**Figure 13.** Numerical calculation flow chart for coupling of multibody dynamics (MBD)-discrete element method (DEM).

The MBD loop is implemented by the software RecurDyn, and the DEM loop is calculated by the software EDEM. In unidirectional coupling, the corresponding movements of the geometries calculated by RecurDyn are transmitted to the corresponding geometries in EDEM. The equivalent particle forces and torques acting on the geometries are calculated by EDEM, and are transmitted to the corresponding geometries in RecurDyn. A coupled MBD–DEM simulation of the industrial-scale inertia cone crusher (model: GYP1200) is calculated with four different shapes of feed simulated particles (50 mm diameter) using BPM, as shown in Figure 14. Figure 14 shows the breakage particles' size becomes smaller as the particles flow along the crushing chamber. The corresponding movements of the crusher were obtained by the software RecurDyn.

### 3.2. Industrial Experiments

To validate the coupled dynamic model, the industrial-scale experiment was conducted at a steel plant in Tangshan, China. The rock material is white marble. Each rock was separated at a 50 mm square screen to get the −50 mm feed material. The displacements and accelerations of the main frame global coordinate frame $(cxyz)_0$ (x-, y-, and z-direction) were collected by displacement and acceleration sensors (model: DH311E). The displacement and acceleration data were acquired using a digital display signal acquisition apparatus (Model: DH5902). The power draw data were measured directly at the crushing process of the control system, and the experimental product size distributions were acquired by square sieves. The GYP1200 type crusher test devices are shown in Figure 15.

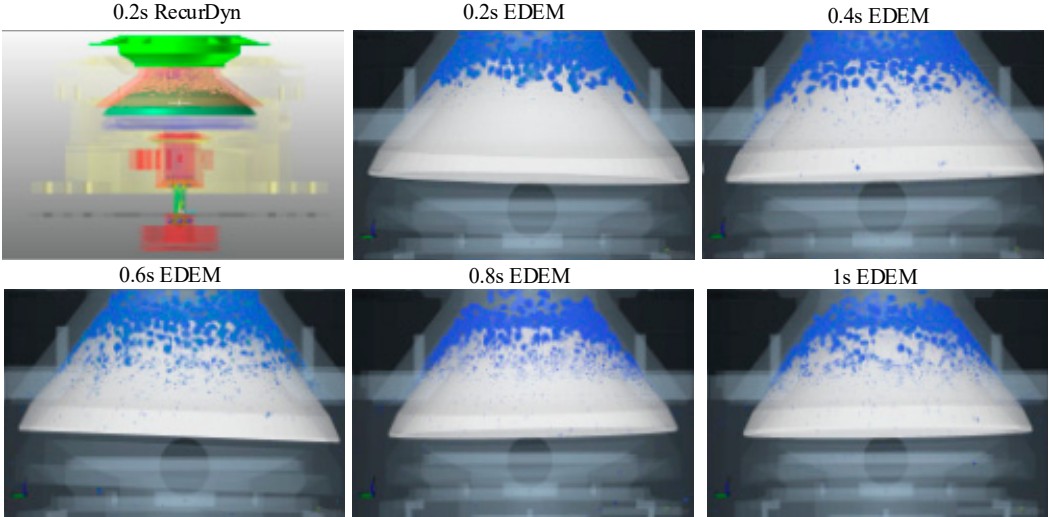

**Figure 14.** Series of pictures from MBD–DEM simulations with the bond cluster.

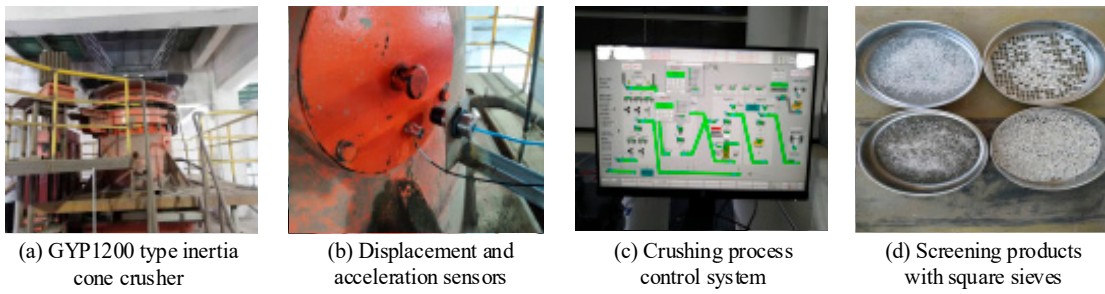

(a) GYP1200 type inertia cone crusher

(b) Displacement and acceleration sensors

(c) Crushing process control system

(d) Screening products with square sieves

**Figure 15.** The GYP1200 type crusher test devices.

The displacements and accelerations of the test point were measured by displacement and acceleration sensors, as shown in Figure 15b. The drive shaft speed setting was carefully calibrated by the crushing process control system, as shown in Figure 15c. The products were split at a series of square sieves in order to get product size distributions, as shown in Figure 15d. The experimental and simulated test results were compared at a 400 and 600 rpm drive speed.

## 4. Results and Discussion

### 4.1. Coupled Model Validation

#### 4.1.1. The Displacements of the Test Point

In an inertia cone crusher, the motion from the eccentric vibrator is transferred via the floating cone to the fixed cone. The vibration amplitudes of displacements hence give a good indication for vibration characteristics of the crusher. The simulated and measured displacements of the test point in the *xyz* axes were displayed for 1 s of operation, as shown in Figure 16. As can be seen, the simulated and measured displacements from Figure 16a,b display a periodic cycle behavior of 6.7 Hz or 10 Hz. This is the same frequency as the speed of the eccentric vibrator.

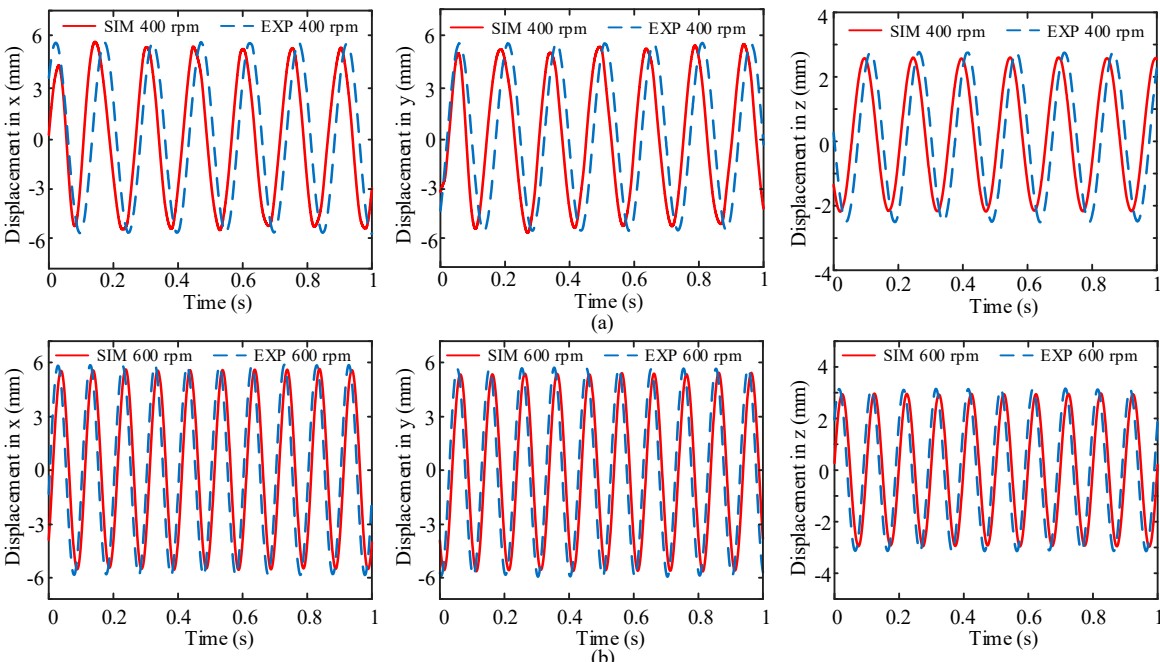

**Figure 16.** Comparison of the displacements in the x-, y-, and z-directions between simulation and experiment for (**a**) 400 rpm and (**b**) 600 rpm.

Figure 16 shows that the simulation results are appropriately consistent with the experiment data. The simulated and experimental mean and the amplitude of displacement is only slightly higher in the 600 rpm than the 400 rpm case.

### 4.1.2. Power Draw

The power draw is a very important performance index as it shows how much energy consumption demands the crusher operation. The output power of the drive shaft is used for several parts: mechanical energy of the inertia cone crusher in nonloaded, crushing rocks, and energy loss due to mechanical damping [38]. The experimental power draw data were measured directly at the crushing process of the control system. The data indicate the input power of the drive motor, hence the value of experimental power multiplied by the motor drive efficiency is the experimental output power of the drive shaft. There is a calculation program for the output power in the RecurDyn software (V9r1, FunctionBay, Seoul, South Korea). The measured and simulated output power of the drive shaft can be seen in Figure 17.

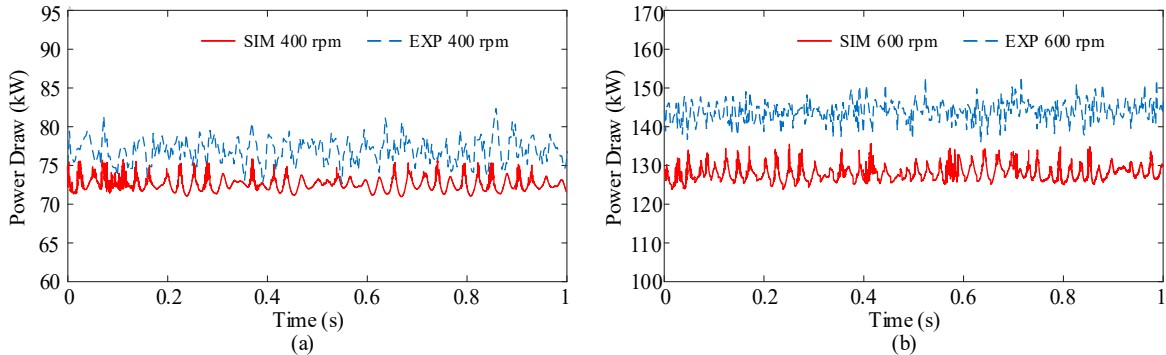

**Figure 17.** The curve displays a comparison between simulated and measured output power for 1 s of operation: (**a**) 400 rpm and (**b**) 600 rpm.

The output power from the simulations and experiments display a fluctuation behavior in the small-scale due to that the particles uniformly distribute in the crusher chamber which is filled with breakage particles. When comparing the mean value of the output power between the simulation and the experiment, the simulation results display a significant difference to the experimental data, especially in the 600 rpm case. It may be that the friction parameters of the joints are not suitable. Even though the output power simulated displays poor correspondence, the trend of change is consistent in a certain error range. There are significant differences between 400 rpm case and 600 rpm case in the output power. This indicates that the output power increases significantly as the speed of the drive shaft increases.

### 4.1.3. Product Size Distribution and Throughput Capacity

In the simulation, when four different shapes of simulated particles are broken, fragments form crushing products together with free metaparticles. Some fragmentation clusters leave the chamber as the coarse particles of the product particle size distribution. The total product size distribution is made up of the surviving fragmentation clusters and free metaparticles. The method applied to obtain the size of the fragmentation clusters and the free metaparticles outside the crushing chamber is based on simulating the operation of product screening. Figure 18 shows the product size distributions between the simulation and the experiment were compared in 400 rpm and 600 rpm case.

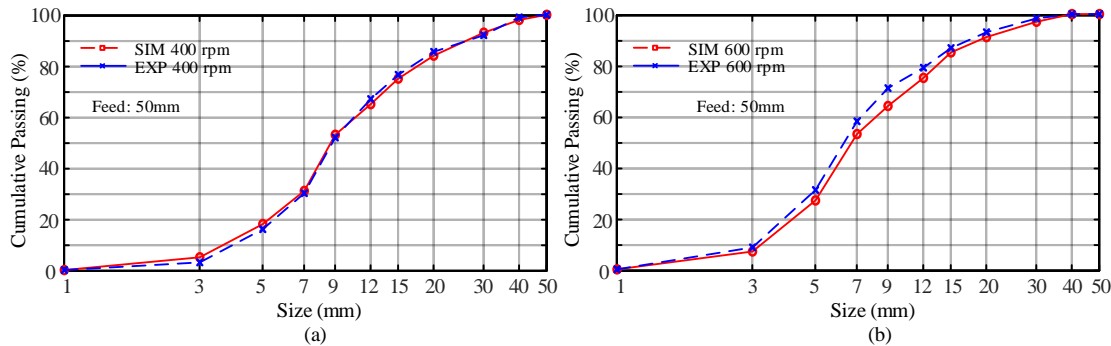

**Figure 18.** Comparison of the simulated and measured product size distribution for (**a**) 400 rpm and (**b**) 600 rpm.

The product particle size index of the GYP1200 type crusher is implemented (the cumulative passing of 12 mm is more than 80%, and the cumulative passing of 1 mm is less than 3%), to avoid severe "excessive combinations". The product size (−1 mm) is less than 2% in the industrial experiments. To compare with the simulation product, we removed the −1 mm experiment product. The experimental product size distribution is set to +1 mm, and as a result, the minimum size limit of the metaparticles from simulation makes it impossible to acquire the product size below that level. The minimum size of the BPM particles used must be adjusted according to the product index. This is the limitation of simulation and cannot be eliminated. Figure 19 shows the simulation results correspond well to the experimental data in the 400 rpm case, whereas in the 600 rpm case, there is a significant difference between the simulation results and the test data. Currently, no hypotheses can explain the problem. It may be that for the 600 rpm case, the experimental product size (−1 mm) significant increases. Furthermore, the simulated particles, which are not enough in simulations, are also the reason for relative differences. Even the simulated 600 rpm case displays poor correspondence and the trend of change is consistent in a certain error range. It indicates the 600 rpm case is finer than the 400 rpm case.

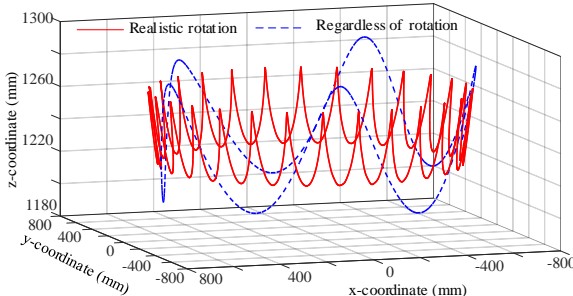

**Figure 19.** Comparison of the trajectory between real and regardless of rotation upon the mantle axis.

The experimental and simulated throughput capacity was calculated by collecting products. In terms of throughput capacity, the simulation results correspond well to the experimental data for the 400 rpm case, whereas there is a significant difference between the simulation results and the test data in the 600 rpm case. Table 5 shows the results from simulations and experiments were summarized. For the primary stages of modeling, if the simulated values are in the reasonable range and show the correct trend corresponding to the experimental data, it should be considered a success.

**Table 5.** Summary of simulated and experimental averaged results.

| Performance | Analysis | SIM 400 rpm | EXP 400 rpm | SIM 600 rpm | EXP 600 rpm |
|---|---|---|---|---|---|
| Amplitude in x (mm) | Mean | 5.25 | 5.46 | 5.57 | 5.74 |
| | Std Dev | 0.121 | 0.099 | 0.136 | 0.159 |
| Power draw (kW) | Mean | 73.8 | 76.8 | 128.7 | 146.2 |
| | Std Dev | 3.12 | 3.97 | 6.79 | 7.81 |
| Throughput (t/h) | Mean | 84.4 | 80.8 | 63.7 | 51.8 |
| Specific energy (kW·h/t) | | 0.87 | 0.95 | 2.02 | 2.82 |

*4.2. Simulated Results*

4.2.1. Trajectory of Fixed Point in the Mantle Surface

Using the simulation, it is possible to evaluate the behavior of the mantle motion and the corresponding to the realistic motion made regarding. In the global coordinate system $(cxyz)_0$, a trajectory of the fixed point in the mantle surface is shown in Figure 19. In the 600 rpm case, it takes 2.7 s for the realistic rotation trajectory of the fixed point to move in one cycle and regardless of whether the rotation takes 0.1 s. Therefore, the rotation of the mantle upon its axis has played a significant role in the process of movement of the mantle and bearing lubrication.

4.2.2. Behavior of the Particle Flow

The behavior of the particle flow in the chamber determines the throughput capacity of the crusher [3]. The simulation can qualitatively evaluate the behavior of the breakage particle flow, for instance, the assumptions of particle dynamics in the compression event. The trajectory streams were presented for two randomly selected breakage particles in Figure 20. The figure shows the simulated particles instantaneously in any of three different states of motion: sliding (downwards), free fall, or squeezing (lifting).

For an inertia cone crusher, the compression event normally consists of several nominal events corresponding to the chamber shape, feed particle size, drive speed, and nutation angle. This figure shows a simulated particle sliding along the circumference when falling along the axis. The repeated lines represent the breakage particle is squeezed (lifted) at each nominal compression event, soon after, the breakage particle slides (free flows) to the next nominal event. The zigzag pattern is more in the

600 rpm case than in the 400 rpm case hence there are more compression events in the 600 rpm case resulting in reduced throughput capacity.

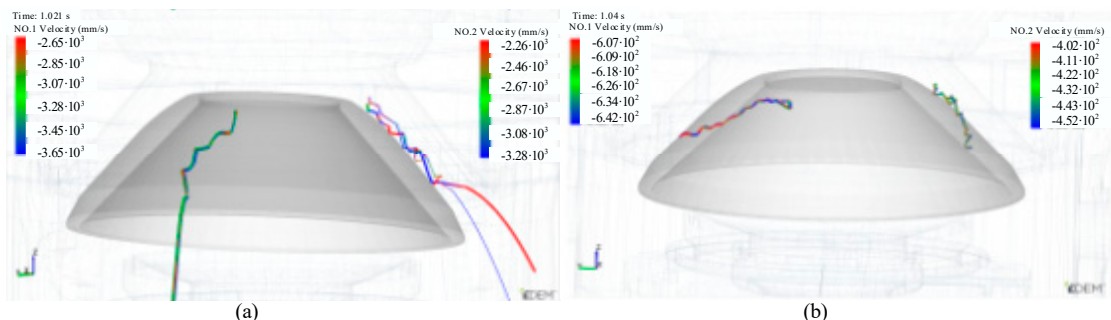

(a)                                                              (b)

**Figure 20.** Path trajectory for two breakage particles in the 400 rpm case (**a**) and two breakage particles from the 600 rpm case (**b**).

### 4.2.3. Crushing Force Distribution in the Concave Surface

The crushing force distribution in the concave surface is one of the main factors determining the linings wear period. As such, the equivalent crushing forces in different areas of the concave were obtained using the simulation. The concave consists of 13 rings in the vertical direction and each ring consists of nine areas. The average normal force of the particle acting on each area denotes the crushing force $F_n$ on the area and the crushing force is proportional to the friction $F_t$. Hence, the crushing force $F_n$ determines the concave wear. For clearly describing that the crushing force acting on different positions changes over time, the $F_n$ acting on $P_{31}$, $P_{51}$, $P_{71}$, $P_{91}$, and $P_{111}$ was selected as a typical location, as shown in Figure 21a. Figure 21b shows the crushing forces acting on the typical locations for 1 s of operation in the simulated 600 rpm case. The crushing forces display a periodic cyclic behavior with the 10 Hz, which is the same as the eccentric vibrator speed.

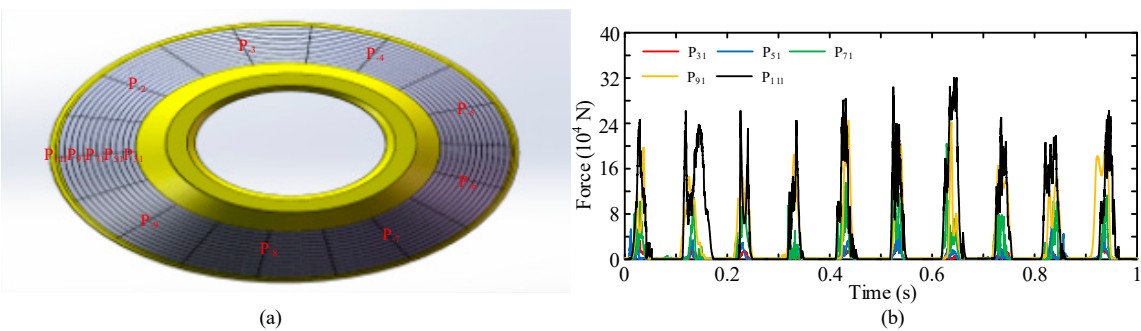

(a)                                                              (b)

**Figure 21.** Locations and numbers of the areas (**a**) $F_n$ acting on $P_{31}$, $P_{51}$, $P_{71}$, $P_{91}$, and $P_{111}$ for 1 s of operation (**b**).

The simulated results quantitatively evaluate the crushing force distribution in the concave, indicating that the magnitude of the average crushing force and friction increase with progressing position numbers. As a result, the rings can be set in different kinds of wear-resistant materials, and it provides novel insight regarding the improvement of the wear period.

### 4.3. Influence of Fixed Cone Mass and Drive Speed on Crusher Performance

#### 4.3.1. Influence on Displacement Amplitude

The effect of drive shaft speed and fixed cone mass on displacement amplitude were considered in this simulation results in Figure 22. Figure 22a shows that the displacement amplitude of fixed cone

for the same conditions is slightly higher with increasing speed. This is because the high speed gives the floating cone enough eccentric inertia force to increase the eccentric distance of the floating cone and eccentric vibrator. Figure 22b shows the relation between displacement amplitude and fixed cone mass for the same conditions, and the displacement amplitude decreases significantly as the fixed cone mass increases. The reason for this is that decreasing the fixed cone mass leads to the movement of the fixed cone easier to change at the same crushing force.

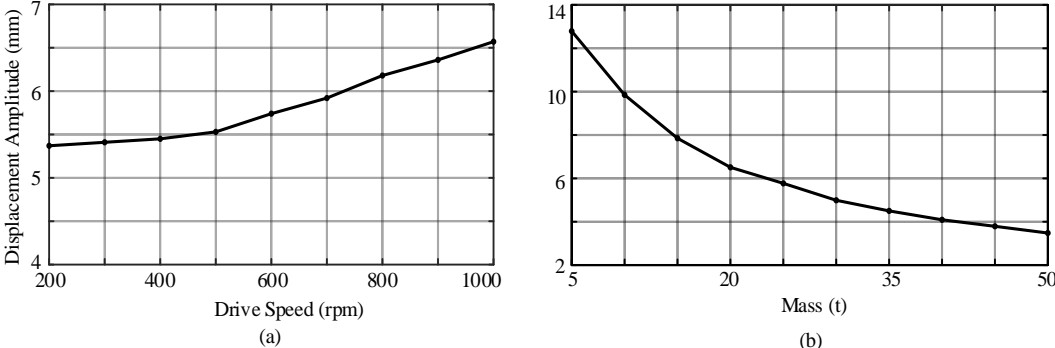

**Figure 22.** The effect of drive speed and fixed cone mass on displacement amplitude as: (**a**) displacement amplitude for different speeds in the fixed cone mass 24 t case; (**b**) displacement amplitude for different masses in the 600 rpm case.

### 4.3.2. Influence on Product Size Distribution

The drive shaft speed could affect the interaction between particles and the crusher chamber. Especially, the product size distribution is affected by the number of nominal compression event and the local compression ratio of the rock particles. The effect of drive shaft speed and fixed cone mass on product size distribution were considered in these simulation results, as shown in Figure 23. Figure 23a illustrates that the size of product is finer with increasing the speed in the same conditions. There is a significant influence of drive shaft speed on product size distribution. The reason is that particles do not have enough time to pass to the next zone and receive more breakage in the high-speed case, besides the high-speed leads to larger the local compression ratio. Figure 23b shows the relation between product size distribution and fixed cone mass for the same conditions, and the size of the product is finer with increasing the fixed cone mass. However, the fixed cone mass affects product size distribution in the range of 10 to 25 t, and there is almost no influence in the range of 25 to 40 t. The reason is that in the condition of large amplitude, the crushing force of the floating cone and eccentric vibrator on the fixed cone is reduced by decreasing the fixed cone mass, leading to fewer particles being selected for breakage.

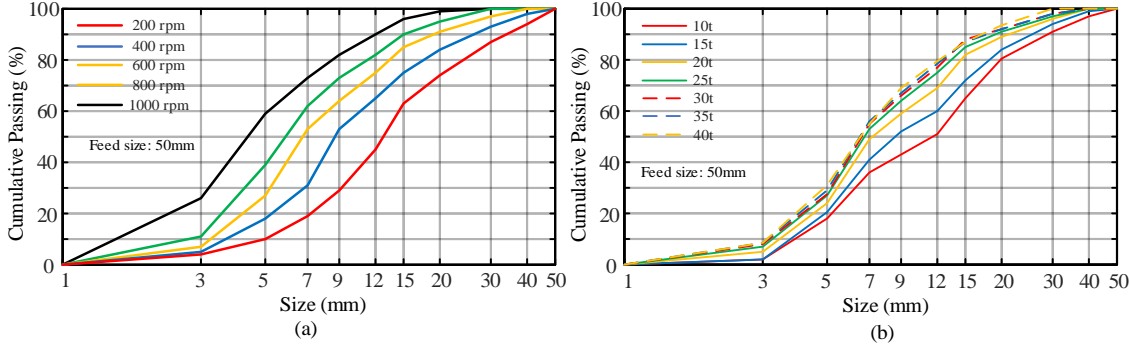

**Figure 23.** The effect of drive speed and fixed cone mass on product size distribution, simulation results as: (**a**) product size distribution for different speeds in the fixed cone mass 24 t case; (**b**) product size distribution for different masses in the 600 rpm case.

### 4.3.3. Influence on Specific Energy Consumption

The effect of drive shaft speed and fixed cone mass on specific energy consumption were considered in these simulation results, as shown in Figure 24. There are significant effects of drive shaft speed on power draw and throughput capacity, thus Figure 24a shows the relation between specific energy consumption and drive shaft speed for the same conditions, and the specific energy consumption increases significantly as the drive shaft speed increases. The reason is why the power draw increases and the throughput capacity declines with increasing speed. The effect of fixed cone mass on the specific energy consumption was considered in this simulation results, as shown in Figure 24b. The figure shows the fixed cone mass significantly affects the specific energy consumption in the range of 5 to 35 t, and there is a little influence in the range of 35 to 50 t. This is because the power draw and throughput capacity change slightly under the condition of small amplitude.

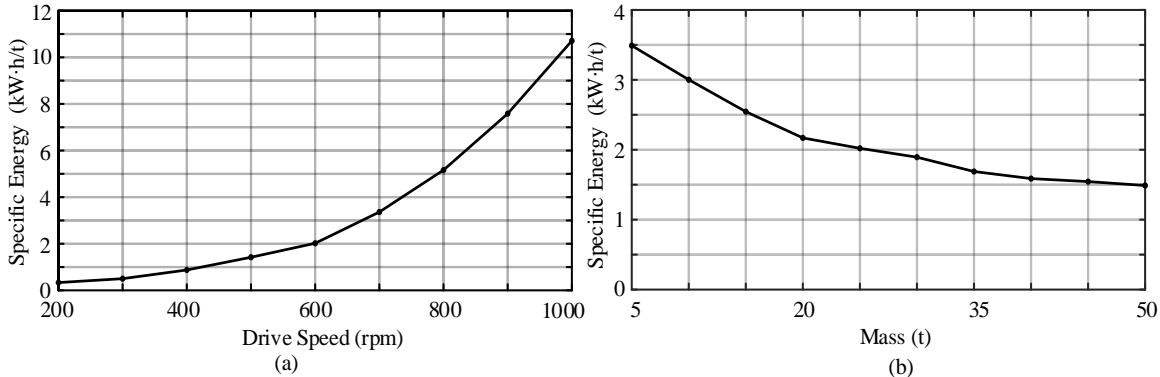

**Figure 24.** The effect of drive speed and fixed cone mass on specific energy consumption simulation results as: (**a**) specific energy consumption for different speeds in the 24 t case; (**b**) specific energy consumption for different masses in the 600 rpm case.

## 5. Conclusions

A coupled MBD–DEM model with bonded particles has been proposed to describe the operation of inertia cone crusher and calibrated against laboratory Brazilian splitting tensile strength tests and single-particle breakage experiments. This model can input the main operating variables in the breakage process directly, for instance, the feed size, the crusher operation parameters, and the geometric parameters to predict the response of equipment dynamically, especially the crusher performance variation with the drive speed and fixed cone mass. In addition, it is a limitation of the work that the errors are relatively high between simulations and tests for the critical compression ratio, and more work is needed to find a different approach to solve this limitation. Using industrial experiments and MBD–DEM simulations, a GYP1200 inertia cone crusher is investigated. It has been validated that the displacement and throughput capacity can be very good agreement however more work is needed to improve the prediction in terms of power draw, especially in product size distribution.

Based on simulation results, the crushing force in the concave vertical direction will be larger from the top to the bottom of the concave. As a result, the wear resistance of the concave can be varied in the vertical direction, in order to improve the wear period. For a given crusher performance, the fixed cone mass can be used in optimizing the design of a crusher, in order to lower manufacturing cost. When the material of feed material changes, the optimal operation parameters (such as drive speed, CSS, and eccentric static moment) vary in time, and the adjustment of drive speed in real-time can result in improving the crusher performance. Future work will prioritize the development of crusher operation optimization systems, the adoption of the Taguchi method for the Design of Experiments, and multiple control variables to further optimize drive speed, CSS, and eccentric static moment.

**Author Contributions:** Conceptualization, J.C. and T.R.; methodology, J.C.; software, J.C.; validation, Z.Z.; formal analysis, D.L.; investigation, J.C. and X.J.; data curation, J.C.; writing—original draft, J.C.; project administration, T.R. All authors have read and agreed to the published version of the manuscript.

**Funding:** This work is supported by the National Key Technology R&D Program of China (grant number.2011BAF15B01).

**Conflicts of Interest:** The authors declare no conflict of interest.

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
