# Peer review of "A Dynamic Model of Inertia Cone Crusher Using the Discrete Element Method and Multi-Body Dynamics Coupling"

_minerals, doi:10.3390/min10100862_

Round 1

Reviewer 1 Report

The work applies, for the first time, MBD coupled to DEM to describe the performance of the inertial roll crushers, using BPM to describe breakage. It is well-written and will certainly be of great interest by readers in the field. However, before the manuscript is accepted, I suggest clarifying the following issues:

The statement “The inertia cone crusher is one of the indispensable equipment in complex ore mineral processing.” in the abstract is unnecessary and does not reflect the practice in crushing, since this type of crusher is only seldom used. I believe it could be replaced by “The cone crusher …”. This could then be followed by “A variant of the cone crusher is the inertia crusher”.

In order to facilitate interpretation, report p-value instead of F value in Table 3.

How were contact parameters presented in Table 4 selected?

It seems that in Figure 18 the authors forcefully assumed that no particles existed below 1 mm. I acknowledge that is the case for the simulated particles, whose meta-particles were coarser than that, but it is imperative that experimental data are presented including such data, otherwise results will be biased.  

In caption of Figure 23, clearly show that the results presented are solely given by simulations.

Minor edits:

  • Line 35: replace therefor by therefore.
  • Line 49: replace industrial by industry.
  • Line 78: the acronym that is most commonly used is HPGR, which stands for high-pressure grinding rolls.
  • Lines 86-76: replace with “…which cannot be described using …”
  • Lines 123 and 127: Start sentence with capital letter.
  • Figure 5 appears to have low resolution.
  • Line 283: sentence “…so that a sufficient property of breakage is achieved…” is not quite clear. Suggest changing to “…so that breakage is properly described…”.
  • Line 293: replace “…different the simulated …” by “…different simulated …”.
  • Line 319: change “The Breakage Simulations are Established Using the Taguchi Method for DOE” to “Breakage Simulations Established Using DOE”
  • Lines 429-430: replace “…and experimental mean the amplitude of …” by “…and experimental mean and the amplitude of …”
  • Line 433: replace “…consumption in the crusher operation…” with “…consumption demands the crusher operation…”
  • Line 500: replace “…instantaneously are in any of three…” with “…instantaneously in any of three…”
  • Line 551: replace “…There is significant the influence…” with “…There is significant influence…”
  • Line 563: replace “…considered in this simulation results,…” with “…considered in these simulation results,…”

Author Response

Response to Reviewer 1 Comments

Thank for the reviewer’s comments concerning our manuscript entitled “A Dynamic Model of Inertia Cone Crusher Using the Discrete Element Method and Multi-Body Dynamics Coupling” (Article: 937361). Those comments are all valuable and very helpful for revising and improving our paper, as well as the important guiding significance to our researches. We have studied comments carefully and have made correction which we hope meet with approval. Revised portion are marked in red in the paper. The main corrections in the paper and the responds to the reviewer’s comments are as following:

Point 1: The statement “The inertia cone crusher is one of the indispensable equipment in complex ore mineral processing.” in the abstract is unnecessary and does not reflect the practice in crushing, since this type of crusher is only seldom used. I believe it could be replaced by “The cone crusher …”. This could then be followed by “A variant of the cone crusher is the inertia crusher”.

Response 1: Considering the reviewer’s suggestion, the first statement in the abstract (The inertia cone crusher is one of the indispensable equipment in complex ore mineral processing) was corrected. It could be replaced by “The cone crusher is one of the indispensable equipment in complex ore mineral processing and a variant of the cone crusher is the inertia crusher”.

Point 2: In order to facilitate interpretation, report p-value instead of F value in Table 3.

Response 2: Considering the reviewer’s suggestion, we replaced F value with P value in Table 3.

Point 3: How were contact parameters presented in Table 4 selected.

Response 3: A slump test was used to estimate the angle of repose of white marble. We took advantage of the material information within the Generic EDEM Material Model (GEMM) Database to obtain contact parameters of white marble in the EDEM simulation by answering 3 questions (the size, the bulk density and the angle of repose) about the bulk material and one question (the wall friction) about the equipment material.

Point 4: It seems that in Figure 18 the authors forcefully assumed that no particles existed below 1 mm. I acknowledge that is the case for the simulated particles, whose meta-particles were coarser than that, but it is imperative that experimental data are presented including such data, otherwise results will be biased.

Response 4: The -1 mm product is less than 2% in the industrial experiments. To compare with the simulation product, we removed the -1 mm experiment product. Due to carelessness, we did not clearly explain the -1mm product was not presented in the industrial experiments. The relevant contents were modified in the paper. The product particle size index of the GYP1200 type crusher is implemented (the cumulative passing of 12 mm is more than 80%, and the cumulative passing of 1 mm is less than 3%), to avoid severe “excessive combinations”. The experimental size distributions are represented as the end +1 mm, and this is done since the minimum size limit of the meta-particles makes it impossible to predict the product size below that level. The minimum size of BPM particle used must be adjusted according to the product index. We acknowledge that results will be biased. In order to avoid the limit, we will consider other ways in further work.

Point 5: In caption of Figure 23, clearly show that the results presented are solely given by simulations.

Response 5: Due to carelessness, we did not clearly show that the results presented are solely given by simulations, in caption of Figure 23. The relevant contents were modified in the paper.

Point 6: Line 35: replace therefor by therefore.

Response 6: The relevant contents were modified in the paper.

Point 7: Line 49: replace industrial by industry.

Response 7: The relevant contents were modified in the paper.

Point 8: Line 78: the acronym that is most commonly used is HPGR, which stands for high-pressure grinding rolls.

Response 8: The relevant contents were modified in the paper.

Point 9: Lines 86-76: replace with “…which cannot be described using …”

Response 9: The relevant contents were modified in the paper.

Point 10: Lines 123 and 127: Start sentence with capital letter.

Response 10: The relevant contents were modified in the paper.

Point 11: Figure 5 appears to have low resolution.

Response 11: The relevant contents were modified in the paper.

Point 12: Line 283: sentence “…so that a sufficient property of breakage is achieved…” is not quite clear. Suggest changing to “…so that breakage is properly described…”.

Response 12: The relevant contents were modified in the paper.

Point 13: Line 293: replace “…different the simulated …” by “…different simulated …”.

Response 13: The relevant contents were modified in the paper.

Point 14: Line 319: change “The Breakage Simulations are Established Using the Taguchi Method for DOE” to “Breakage Simulations Established Using DOE”.

Response 14: The relevant contents were modified in the paper.

Point 15: Lines 429-430: replace “…and experimental mean the amplitude of …” by “…and experimental mean and the amplitude of …”.

Response 15: The relevant contents were modified in the paper.

Point 16: Line 433: replace “…consumption in the crusher operation…” with “…consumption demands the crusher operation…”.

Response 16: The relevant contents were modified in the paper.

Point 17: Line 500: replace “…instantaneously are in any of three…” with “…instantaneously in any of three…”.

Response 17: The relevant contents were modified in the paper.

Point 18: Line 551: replace “…There is significant the influence…” with “…There is significant influence…”.

Response 18: The relevant contents were modified in the paper.

Point 19: Line 563: replace “…considered in this simulation results,” with “…considered in these simulation results,”.

Response 19: The relevant contents were modified in the paper.

We tried our best to improve the manuscript and made some changes in the manuscript. These changes will not influence the content and framework of the paper. We appreciate for Editors/Reviewers’ warm work earnestly, and hope that the correction will meet with approval. Once again, thank you very much for your comments and suggestions.

Jiayuan Cheng

Yanshan University

Address

Tel: +86-15903330972

Reviewer 2 Report

Overall, a good effort in a difficult area.

Most of my comments relate to minor corrections of English expression.

A few technical terms are not in common use in processing and deserve some definition to help the reader.

The description of the offset weight and how it is driven is not sufficiently clear. Please add some detail.

Line 90 parameter estimates for

105 broken particles

113 and is called

123 S

154 adding a sentence to define "holonomic constraint" will be helpful to many readers.

168 can experience this

perhaps you should mention that this condition should be avoided

The Taguchi method  deserves a reference

2.4 The BPM method is not in common use because it requires substantial ore calibration - which you have under taken.

The method referenced in the Cleary et al reference is more more general and you might consider it in further work. It is not required for this paper.

277 Which error? Between estimated and measured progeny sizing?

280 Particle Packing Cluster

Figure 9 for six Brazilian tests

Reference for RecurDyn?

407 square mesh sizes

461 reference for GPY 1200

486 Using the simulation it is possible

Author Response

esponse to Reviewer 2 Comments

Thank for the reviewer’s comments concerning our manuscript entitled “A Dynamic Model of Inertia Cone Crusher Using the Discrete Element Method and Multi-Body Dynamics Coupling” (Article: 937361). Those comments are all valuable and very helpful for revising and improving our paper, as well as the important guiding significance to our researches. We have studied comments carefully and have made correction which we hope meet with approval. Revised portion are marked in red in the paper. The main corrections in the paper and the responds to the reviewer’s comments are as following:

Point 1: A few technical terms are not in common use in processing and deserve some definition to help the reader.

Response 1: Considering the reviewer’s suggestion, a few technical terms were defined to help the reader. The relevant contents were modified in the paper.

Point 2: The description of the offset weight and how it is driven is not sufficiently clear. Please add some detail.

Response 2: In terms of the offset weight, we changed the weight of the crusher by setting the parameters of the software RecurDyn. In the industrial experiment, we did not change the weight of the crusher. The content of the section 4.3 is based on the simulation results. In the industrial experiment, we used a variable-frequency motor to change the speed of the drive shaft. In the simulation, we changed the speed by setting the motion parameters of revolute joint.

Point 3: Line 90 parameter estimates for.

Response 3: The relevant contents were modified in the paper.

Point 4: Line 105 broken particles.

Response 4: The relevant contents were modified in the paper.

Point 5: Line 113 and is called

Response 5: The relevant contents were modified in the paper.

Point 6: Line 123 S.

Response 6: The relevant contents were modified in the paper.

Point 7: Line 154 adding a sentence to define "holonomic constraint" will be helpful to many readers.

Response 7: The holonomic constraint is defined as a set of constraint equations with only displacement coordinates. The relevant contents were modified in the paper.

Point 8: Line 168 can experience this. Perhaps you should mention that this condition should be avoided.

Response 8: While the inertia cone crusher is working normally, the mantle and the concave will eventually come into contact or collide and will exert forces between them. Because the inertia cone crusher has nonlinear contact problem, nonlinear contact model needs to be studied. Due to carelessness, we did not clearly explain the necessity of nonlinear contact model. The relevant contents were modified in the paper.

Point 9: The Taguchi method deserves a reference.

Response 9: Considering the reviewer’s suggestion, Taguchi’s research was added to the references of the paper.

Point 10: 2.4 The BPM method is not in common use because it requires substantial ore calibration which you have under taken. The method referenced in the Cleary et al reference is more general and you might consider it in further work. It is not required for this paper.

Response 10: Because there are some limitations in the BPM method and the number of ore calibration tests is not enough, we will consider the Cleary et al reference in further work.

Point 11: Line 277 Which error? Between estimated and measured progeny sizing?

Response 11: In terms of the critical force and the compression ratio, there are differences between simulation and test. By minimizing the differences, we can find the optimum parameters for the BPM model.

Point 12: Line 280 Particle Packing Cluster.

Response 12: The relevant contents were modified in the paper.

Point 13: Figure 9 for six Brazilian tests.

Response 13: The relevant contents were modified in the paper.

Point 14: Reference for RecurDyn?

Response 14: The software RecurDyn is developed and designed by FunctionBay (South Korea) for MBD simulation.

Point 15: Line 407 square mesh sizes.

Response 15: The square mesh sizes: 1, 3, 5, 7, 9, 12, 15, 20, 30, 40 (mm).

Point 16: Line 461 reference for GPY 1200.

Response 16: The GYP1200 type inertia cone crusher is designed and manufactured by Beijing Kaite company. The geometric parameters of the crusher were presented in Table 1. Beijing Kaite company provided the product particle size index of the GYP1200 type crusher.

Point 17: Line 486 Using the simulation it is possible.

Response 17: The relevant contents were modified in the paper.

We tried our best to improve the manuscript and made some changes in the manuscript. These changes will not influence the content and framework of the paper. We appreciate for Editors/Reviewers’ warm work earnestly, and hope that the correction will meet with approval. Once again, thank you very much for your comments and suggestions.

Jiayuan Cheng

Yanshan University

Address

Tel: +86-15903330972
